# Multi-Layer Hidden Markov Model Based Intrusion Detection System

**Wondimu K. Zegeye \*, Richard A. Dean and Farzad Moazzami**

Department of Electrical and Computer Engineering, Morgan State University, Baltimore, MD 21251, USA; richard.dean@morgan.edu (R.A.D.); farzad.moazzami@morgan.edu (F.M.)

**\*** Correspondence: wondimuk@gmail.com

**Abstract:** The all IP nature of the next generation (5G) networks is going to open a lot of doors for new vulnerabilities which are going to be challenging in preventing the risk associated with them. Majority of these vulnerabilities might be impossible to detect with simple networking traffic monitoring tools. Intrusion Detection Systems (IDS) which rely on machine learning and artificial intelligence can significantly improve network defense against intruders. This technology can be trained to learn and identify uncommon patterns in massive volume of traffic and notify, using such as alert flags, system administrators for additional investigation. This paper proposes an IDS design which makes use of machine learning algorithms such as Hidden Markov Model (HMM) using a multi-layer approach. This approach has been developed and verified to resolve the common flaws in the application of HMM to IDS commonly referred as the curse of dimensionality. It factors a huge problem of immense dimensionality to a discrete set of manageable and reliable elements. The multi-layer approach can be expanded beyond 2 layers to capture multi-phase attacks over longer spans of time. A pyramid of HMMs can resolve disparate digital events and signatures across protocols and platforms to actionable information where lower layers identify discrete events (such as network scan) and higher layers new states which are the result of multi-phase events of the lower layers. The concepts of this novel approach have been developed but the full potential has not been demonstrated.

**Keywords:** Intrusion Detection System (IDS); Hidden Markov Model (HMM); multi-stage attacks

## 1. Introduction

Intrusion Detection Systems have been the subject of a lot of research both in academia and industry in the past few decades as the interest in information security has been growing rapidly. The National Institute of Standards and Technology (NIST) defines intrusion detection as "the process of monitoring the events occurring in a computer system or network and analyzing them for signs of intrusions, defined as attempts to compromise the confidentiality, integrity, availability, or to bypass the security mechanisms of a computer or network." A system which addresses or automates intrusion detection is referred as Intrusion Detection System (IDS) [1].

Intrusion detection systems come in different flavors and approaches. Based on their points of placement, they can be categorized into network-based intrusion detection system (NIDS) and host-based intrusion detection system (HIDS). A network intrusion detection system (NIDS) is placed at a strategic point in the network such that packets traversing a particular network link can be monitored. NIDSs monitor a given network interface by placing it in promiscuous mode. This will help the IDS in hiding its existence from network attackers while performing the task of network traffic monitoring. On the other hand, Host-based IDSs monitor and reside in individual host machines.

HIDSs operate by monitoring and analyzing the host system internals such as operating system calls and file systems. In the same way as NIDS, it can also monitor the network interface of the host.

The techniques employed by modern day IDSs to gather and analyze data are extremely diverse. However, those techniques have common basic features in their structures: a detection module which collects data that possibly contain evidence of intrusion and an analysis engine which processes this data to identify intrusive activity. Those analysis engines mainly use two techniques of analysis: anomaly detection and misuse detection.

The intrinsic nature of misuse detection revolves around the use of expert system which is capable of identifying intrusions mainly based on a preordained knowledge base. Consequently, misuse structures are able to reach very high levels of accuracy in identifying even very subtle intrusions which might be represented in their knowledge base; similarly, if this expert know-how base is developed carefully, misuse systems produce a minimum number of false positives [2].

Unfortunately these architectures are less fortunate due to the fact that a misuse detection system is incapable of detecting intrusions that are not represented in its know-how base. Subtle versions of known attacks may additionally affect the evaluation if a misuse system is not always well constructed. Therefore, the efficiency of the system is highly dependent on the thorough and accurate creation of this information base, undertaking that calls for human expertise involvement, thus, the need to develop anomaly detection methods.

A wide variety of strategies have been explored to detect anomalous events from normal ones including neural networks, statistical modeling and Hidden Markov Models to name a few. Those approaches rely on the same principle. At first, a baseline model that is a representative of normal system behavior against which anomalous events can be distinguished is established. When an event indicates anomalous activity, as compared with the baseline, it is considered as malicious. This system characterization can be used to identify anomalous traffic from normal traffic.

One of the very attractive features of anomaly based intrusion detection systems is their capability to identify previously unseen attacks. The baseline model in this case is usually automated and it does not require both human interference and the knowledge base. The aftermath of this approach is that the detection system may fail to detect even well-known attacks if they are crafted not to be substantially different from the normal behavior established by the system.

Currently, more than fifty percent of the web traffic is encrypted-both normal and malicious. The volume of encrypted traffic is expanding even more which creates confusions and challenges for security teams trying to monitor and identify malicious network traffic. The main goal of encryption is to enhance network security but at the same time it provides intruders the power to hide command-and-control (C2) activity giving them enough time to launch attack and cause damage. To keep up with the intruders, security teams need to include additional automation and modern tools that are developed using machine learning and artificial intelligence to supplement threat detection, prevention and remediation [3].

More enterprises are now exploring machine learning and artificial intelligence to prevail over the effect of encryption and decrease adversaries' time. These advanced concepts have the capability to keep up their performance without humans having to clarify precisely the way to accomplish the tasks that they are provided. Unusual patterns of web traffic that can indicate malicious activity can be automatically detected as these advanced systems can, overtime, "learn" by themselves.

To automatically detect "known-known" threats, the types of attacks that have been known previously, machine learning plays a significant role. But its main advantage in monitoring encrypted web traffic is due to the fact that it is capable of detecting "known-unknown" threats (previously unknown distinct form of known threats, malware subdivision, or similar new threats) and "unknown-unknown" (net-new malware) threats. Those technologies automatically alert potential attacks to network administrators as they can learn to identify unusual patterns in massive volumes of encrypted web traffic.

Those automatic alerts are very important in organizations where there is a lack of knowledgeable personnel in the enhancement of security defenses. Intelligent and automated tools using machine learning and artificial intelligence can help security teams fill the gaps in skills and resources, making them more practical at recognizing and responding to both well-known and prominent threats.

Several techniques of artificial intelligence (AI) have been explored in the path towards developing IDSs, for example, fuzzy logic, artificial neural networks (ANNs) and genetic algorithms (GA). In addition, hybrid intelligent IDSs, such as evolutionary neural network (ENN) and evolutionary fuzzy neural networks (EFuNN)—based IDSs, are also used. This work proposes the potential application of the Hidden Markov Model (HMM) for intrusion detection which is capable of providing finer-grained characterization of network traffic using a multi-layer approach.

The current implementations of HMMs for IDS are mainly based on a single HMM which will be trained for any incoming network traffic to identify anomalous and normal traffic during testing.

Other HMM based IDS implementations rely on multi HMM profiles where each of the HMMs are trained for a specific application type traffic and posterior probabilities are used to select network applications using only packet-level information that remain unchanged and observable after encryption such as packet size and packet arrival time [4]. This approach, even if it factors based on application layer traffic, it considers a limited number of features and is unable to detect multistage attacks which can be crafted to look like normal traffic. Similarly, the work in Reference [5] applied several pre-processing techniques on the dataset considered to implement a multi class system (MCS) HMM based IDS.

Besides to the capability of multi-layer HMM design to detect multi-stage attacks, the IDS data analysis engine of the approach proposed here, on the other hand, uses several features where a dimension reduction technique will be applied to extract only important features. A single layer HMM which attempts to detect multi-stage attacks from a dataset made up of several steps of an attack trace is implemented in Reference [6]. Recent HMM based implementations, such as in Reference [7], are based on a database of HMM templates corresponding to each attack and eventually used to detect multiple multi-stage attacks.

HMMs use statistical learning algorithms that suffer in cost exponentially as the volume of data grows. This aspect is commonly referred as the curse of dimensionality [8]. The HMMs tend to fail, more often than not, on a large dimension state space. Considering a single HMM based IDS, as the incoming network traffic will have a large hidden state space, it will fall a victim to this curse of dimensionality.

The solution to those problems involves the principle of decomposition. Decomposition technique [9] comprises of the following steps: the algorithm of decomposition is applied to lower levels to construct each individual HMMs, representing partial solutions, which will be combined to form the final global solution.

The multi-layer approach proposed in this work has been developed and verified to resolve the common flaws in the application of HMM to IDS commonly referred as the curse of dimensionality. It factors a huge problem of immense dimensionality to a discrete set of manageable and reliable elements. The multi-layer approach can be expanded beyond 2 layers to capture multi-phase attacks over longer spans of time. A pyramid of HMMs can resolve disparate digital events and signatures across protocols and platforms to actionable information where lower layers identify discrete events (such as network scan) and higher layers new states which are the result of multi-phase events of the lower layers. The concepts of this novel approach have been developed but the full potential has not been demonstrated as will be described in the next sections.

## 2. Design Approach

A challenge in applying the Markov model to intrusion detection systems is the lack of a standard method for the translation of the observed network packet data into meaningful Markov model.

The first step towards building an IDS based on multiple layers of Hidden Markov Models involves processing network traffic into meaningful observations.

The IDS designed in this work is as specified in flow chart of Figure 1. It starts by data processing of the captured Wireshark file using feature generation, feature selection among those generated features or creation of new features by combining the generated features, use machine learning algorithm for dimension reduction and finally, apply vector quantization techniques to create meaningful observations for the HMMs. The details of the layered HMM design is discussed in Section 2.5 and the corresponding model structure is as shown in Figure 3.

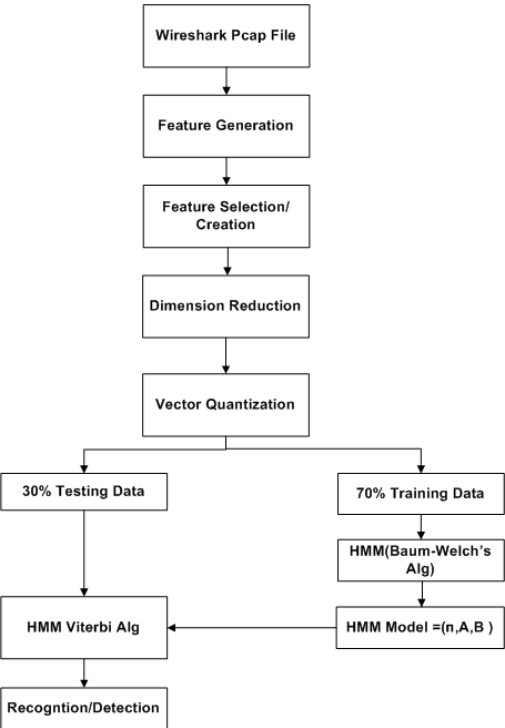

**Figure 1.** Flow Chart of the IDS model including the data processing.

## 2.1. CICIDS2017 Data Processing

Datasets such as DARPA98-99(Lincoln Laboratory 1998–1999), KDD'99 (University of California, Irvine 1998–1999), DEFCON (The Shmoo Group, 2000–2002) UMASS (University of Massachusetts 2011), ISC2012 (University of New Brunswick 2012) and ADFA'13 (University of New South Wales 2013), to name a few, have been used by several researchers to validate their IDS's designs and approaches. Those datasets have shortcomings associated with them. For example, DARPA98-99 and KDD'99 are outdated and unreliable as several flows are redundant; NSL-KDD has been used to replace KDD'99 by removing the significant redundancy. The work in Reference [10] further analyzed the KDD'99 and presented new proofs to these existing inconsistencies. CAIDA (Center of Applied Internet Data Analysis 2002–2016) traffic payloads are heavily anonymized. ADFA'13 lacks attack diversity and variety. Besides to these popular datasets among various researchers, several others are also explored in Reference [11].

Most of the datasets are also simulate attacks which are multi-stage. The CICIDS2017 dataset is designed to cover what are commonly referred as the eleven criteria for IDS dataset evaluation framework (Network, traffic, Label, Interaction, capture, Protocols, Attack diversity, anonymization, heterogeneity, features and metadata). The details of this dataset compared to the commonly available IDS evaluation datasets is presented in Reference [12].

For testing the modeled IDS in this work, a dataset is prepared from the CICIDS2017 dataset. This dataset covers eleven criteria which are necessary in building a reliable benchmark dataset. It

contains very common attacks such as XSS, Port scan, Infiltration, Brute Force, SQL Injection, Botnet DoS and DDoS. The dataset is made up of 81 features including the label and it is publicly available in Canadian Institute for Cybersecurity website [13]. Some of these features are directly extracted while others are calculated for both normal (benign) and anomalous flows using the CICFlowMeter [14].

The CICIDS2017 dataset comprises 3.1 million flow records and it covers five days. It includes the following labeled attack types: Benign, DoS Hulk, Port Scan, DDoS, DoS, FTP Patator, SSH Patator, DoS Slow Loris, DoS Slow HTTP Test, Botnet, Web Attack: Brute Force, Web Attack: Infiltration, Web Attack: SQL Injection and Heartbleed. The portion of the "Thursday-WorkingHours-Morning-webattack" is used to create the training and testing dataset which constitutes "BENIGN", "SSH Patator" and "Web Attack-Brute Force" traffic.

### 2.2. Feature Selection and Creation

Feature selection involves choosing a subset of features from the initial available features whereas feature creation is a process of constructing new features and it is usually performed after feature selection process. Feature selection takes a subset of features (M) from the original set of features (N) where M < N.

To build a robust and high performance IDS, the features created or constructed from the subset of selected features could follow a knowledge-based approach. Other approaches which can be applied to construct new features are data-driven, hypothesis-driven and hybrid [15,16].

From Features discarded include Source port. This is due to the fact that as it is part of the Categorical encoding is applied on the following features (Flow_ID, Source_IP, Destination_IP), resulting in numerical values. In addition, a new feature (label) which identifies the traffic type such as BENIGN, SSH-patator and web-attack-bruteforce is added. The values corresponding to this new feature are also categorically encoded.

### 2.3. Feature Extraction/Dimension Reduction

On the other hand, dimension reduction is one form of transformation whereby a new set of features is extracted. This feature extraction process extracts a set of new features from the initial features through some functional mapping [17].

Feature extraction, also known as dimension reduction, is applied after feature selection and feature creation procedures on the original set of features. Its goal is to extract a set of new features through some functional mapping. If we initially have $n$ features (or attributes), $A_1, A_2, \ldots, A_n$, after feature selection and creation, feature extraction results in a new set of features, $B_1, B_2, \ldots, B_m$ ($m < n$) where $B_i = F_i(A_1, A_2, \ldots, A_n)$ and $F_i$ is a mapping function.

PCA is a classic technique which is used to compute a linear transformation by mapping data from a high dimensional space to a lower dimension. The original $n$ features are replaced by another set of $m$ features that are computed from a linear combination of these initial features.

Principal Component Analysis is used to compute a linear transformation by mapping data from a high dimensional space to a lower dimension. The first principal component contributes the highest variance in the original dataset and so on. Therefore, in the dimension reduction process, the last few components can be discarded as it only results in minimal loss of the information value [18].

The main goals of PCA:

1. Extract the maximum variation in the data.
2. Reduce the size of the data by keeping only the significant information.
3. Make the representation of the data simple.
4. Analyze the structure of the variables (features) and observations.

Generally speaking, PCA provides a framework for minimizing data dimensionality by identifying principal components, linear combinations of variables, which represent the maximum variation in the data. Principal axes linearly fit the original data so the first principal axis minimizes the

sum of squares for all observational values and maximally reduces residual variation. Each subsequent principal axis maximally accounts for variation in residual data and acts as the line of best fit directionally orthogonal to previously defined axes. Principal components represent the correlation between variables and the corresponding principal axes. Conceptually, PCA is a greedy algorithm fitting each axis to the data while conditioning upon all previous axes definitions. Principal components project the original data onto these axes, where these axes are ordered such that Principal Component 1 ($PC_1$) accounts for the most variation, followed by $PC_2, \ldots, PC_p$ for $p$ variables (dimensions).

Steps to compute PCA using Eigen Value Decomposition (EVD):

1. From each of the dimensions subtract the mean.
2. Determine the covariance matrix.
3. Determine the eigenvalues and eigenvectors of the calculated covariance matrix.
4. Reduce dimensionality and construct a feature vector.
5. Determine the new data.

The PCA procedure used applies Singular Value Decomposition (SVD) instead of Eigenvalue Decomposition (EVD) [19]. SVD is numerically more stable as it avoids the computation of the covariance matrix which is an expensive operation.

**Singular Value Decomposition (SVD) for Principal Component Analysis (PCA)**

Any matrix X of dimension $N \times d$ can be uniquely written as $X = U \times \Sigma \times V^T$
Where

- r is the rank of matrix $X$ (i.e., the number of linearly independent vectors in the matrix)
- U is a column- orthonormal matrix of dimension $N \times d$
- $\Sigma$ is a diagonal matrix of dimension $N \times d$ where $\sigma_i$'s (the singular values) are sorted in descending order across the diagonal.
- V is a column- orthonormal matrix of dimension $d \times d$.

Given a data matrix X, the PCA computation using SVD is as follows

- For $X^T X$, a rank $r$ $(N \geq d \Rightarrow r \leq d)$, square, symmetric $N \times N$ matrix

    - $\{\hat{v}_1, \hat{v}_2, \ldots, \hat{v}_r\}$ is the set of orthonormal $d \times 1$ Eigenvectors with Eigenvalues $\{\lambda_1, \lambda_2, \ldots, \lambda_r\}$

- The principal components of $X$ are the eigenvectors of $X^T X$
- $\sigma_i = \sqrt{\lambda_i}$ are positive real and termed singular values
- $\{\hat{u}_1, \hat{u}_2, \ldots, \hat{u}_r\}$ is the set of orthonormal $N \times 1$ vectors defined by

    - $\hat{u}_i = \frac{1}{\sigma_i} X \hat{v}_i$
    - $X \hat{v}_i = \sigma_i \hat{u}_i$ (the "value" form of SVD) where $\|X\hat{v}_i\| = \sigma_i$

- $\Sigma$ is $N \times d$ and diagonal

    - $\sigma_i$ are called *singular* values of $X$ . It is assumed that $\sigma_1 \geq \sigma_2 \geq \ldots \geq \sigma_r \geq 0$ (rank ordered).

For N > (r = d), the bottom N-r rows of $\Sigma$ are all zeros which will be removed and the first r rows of $\Sigma$ and the first r columns of U will be kept which results in the decomposition shown in Figure 2.

**PCA and SVD relation**

Let $X = U\Sigma V^T$ be the SVD of matrix X and $C = \frac{1}{N-1} X^T X$ be its covariance matrix of dimension $d$ x $d$. The Eigenvalues of C are the same as the right singular vectors of X.

This can be shown with the following proof:

$$X^T X = V\Sigma U^T U\Sigma V^T = V\Sigma\Sigma V^T = V\Sigma^2 V^T \tag{1}$$

$$C = V\frac{\Sigma^2}{N-1}V^T \tag{2}$$

As $C$ is symmetric, hence $C = V\Lambda V^T$. As a result, the eigenvectors of the covariance matrix are the same as the matrix $V$ (right singular vectors) and the eigenvalues of $C$ can be determined from the singular values $\lambda_i = \frac{\sigma_i^2}{N-1}$

PCA using EVD and SVD can be summarized as follows:

*Objective*: project the original data matrix X using the largest $m$ principal components, $V = [v_1, \ldots, v_m]$.

1. Zero mean the columns of X.
2. Apply PCA and SVD to find the principal components of X.

**PCA:**

a. Determine the covariance matrix, $C = \frac{1}{N-1}X^T X$.
b. V corresponds to the Eigenvectors of C.

**SVD:**

a. Determine the SVD of $X = U\Sigma V^T$
b. V corresponds to the right singular vectors.

3. Project the data in an $m$ dimensional space: $Y = XV$

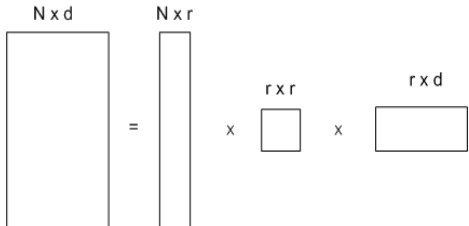

**Figure 2.** Decomposition of matrix X.

To perform dimension reduction and form a feature vector using PCA, order the eigenvalues from the highest to lowest by value. This ordering places the components in order of significance to the variance of the original data matrix. Then we can discard components of less significance. If a number of eigenvalues are small, we can keep most of the information we need and loss a little or noise. We can reduce the dimension of the original data.

E.g. we have data of $d$ dimensions and we choose only the first $r$ eigenvectors.

$$\frac{\sum_{i=1}^{r}\lambda_i}{\sum_{i=1}^{d}\lambda_i} = \frac{\lambda_1 + \lambda_2 + \ldots + \lambda_r}{\lambda_1 + \lambda_2 + \ldots + \lambda_r + \ldots + \lambda_d} \tag{3}$$

$$\text{Feature Vector} = (\lambda_1\,\lambda_2\,\lambda_3\ldots\lambda_r) \tag{4}$$

Since each PC is orthogonal, each component independently accounts for data variability and the Percent of Total Variation Explained (PTV) is cumulative. PCA offers as many principal components as variables in order to explain all variability in the data. However, only a subset of these principal components is notably informative. Since variability is shifted into leading PCs, many of the

remaining PCs account for little variation and can be disregarded to retain maximal variability with reduced dimensionality.

For example, if 99% total variation should be retained in the model for *d* dimensional data, the first *r* principal components should be kept such that

$$PTV = \frac{\sum_{k=1}^{r} \lambda_k}{\sum_{k=1}^{d} \lambda_k} \geq 0.99$$

PTV acts as the signal to noise ratio, which flattens with additional components. Typically, the number of informative components *r* is chosen using one of three methods: (1) Kaiser's eigenvalue > 1; (2) Cattell's scree plot; or (3) Bartlett test of sphericity.

The amount of variation in redundant data decreases from the first principal component onwards. There are several methods to compute the cut off value for retaining the sufficient number of principal components out of all *p* components. In this work, we use the Cattell's scree plot, which plots the eigenvalues in decreasing order [20]. The number of principal components to be kept is determined by the elbow where the curve becomes asymptotic and additional components provide little information. Another method is the Kaiser criterion Kaiser [21] which retains only factors with eigenvalues >1.

*2.4. Vector Quantization*

Vector quantization (VQ) historically has been used in signal representation to produce feature vector sequence. One of the applications of K-Means is vector quantization, thus information theory terminologies used in VQ are commonly applied. For example, the "code book" represents the set of cluster centroids and "code words" represent the individual cluster centroids. The codebook maps the cluster indexes, also known as "code," to the centroids. A basic VQ can be achieved using K-Means clustering with a goal of finding encoding of vectors which minimizes the expected distortion.

2.4.1. Normalization

Before applying VQ on the data matrix resulting from the PCA, the normalization process is applied on the original data matrix preceding the PCA steps. Normalization is helpful in generating effective results as it standardizes the features of the dataset by giving them equal weights. In doing so, noisy or redundant objects will be removed resulting in a dataset which is more reliable and viable, it improves the accuracy. Normalization can be performed using several methods such as Min-Max, Z-Score and Decimal Scaling, to name a few. In a dataset, such as the one used in this work, where among attributes there is a high degree of variation the utilization of another type of normalization known as log-normalization is very convenient [22]. The notations and steps for applying this normalization are described below.

*Notation:*

$x_{ij}$—the initial value in row i and column j of the data matrix
$b_{ij}$—the adjusted value which replaces $x_{ij}$
The transformation below is a generalized procedure that (a) tends to preserve the original order of magnitudes in the data and (b) results in values of zero when the initial value was zero.

*Given:*

$Min(x)$ is the smallest non-zero value in the data
$Int(x)$ is a function that truncates x to an integer by dropping digits after decimal point
$c$ = order of magnitude constant = $Int(log(Min(x)))$
$d$ = decimal constant = $log^{-1}(c)$
Then the transformation is

$$b_{ij} = log(x_{ij} + d) - c$$

A small number must be added to all data points if the dataset contains zeros before applying the log-transformation. For a data set where the smallest non-zero value is 1, the above transformation will be simplified to

$$b_{ij} = log(x_{ij} + 1)$$

### 2.4.2. Clustering

Once the PCA procedure is applied to the initial data, it results in the mapping of the data to a new feature space using the principal components. In the newly constructed feature space, VQ (clustering) is achieved by applying K-Means algorithm.

K-Means objective function [23]:

- Let $\mu_1, \ldots \mu_k$ be the K cluster centroid (means)
- Let $r_{nk} \in \{0, 1\}$ denotes whether point $x_n$ belongs to cluster $k$.
- It minimizes the total sum of distances of each points from their cluster centers (total distortion)

$$J(\mu, r) = \sum_{n=1}^{N} \sum_{k=1}^{K} r_{nk} \|x_n - \mu_k\|^2$$

The steps followed by this algorithm are as follows:

1. Input: N examples $\{x_1, x_2, \ldots, x_n\}; (x_n \in \mathbb{R}^D)$
2. Initialization: K cluster centers $\mu_1, \ldots \mu_k$. K can be initialized:

   - Randomly initialized anywhere in $\mathbb{R}^D$ or
   - Randomly take any K examples as the initial cluster centers.

3. Iteration:

   - Assign each of example $x_n$ to its closest cluster center

   $$C_k = \{n : k = \arg \min_k x_n - \mu_k^2\}$$
   ($C_k$ corresponds to the set of examples closest to $\mu_k$)

   - Re-calculate the new cluster centers $\mu_k$ (mean/centroid of the set $C_k$)

   $$\mu_k = \frac{1}{|C_k|} \sum_{n \in C_k} x_n$$

   - Repeat until convergence is achieved: A simple convergence criteria can be considered as the cluster centers do not move anymore.

To put it in a nut shell, K-Means Clustering is an algorithm which attempts to find groups in a given number of observations or data. Each element of the vector $\mu$ refers to the sample mean of its corresponding cluster, $x$ refers to each of the examples and $C$ contains the assigned class labels [24].

The optimal number of clusters is determined using the Elbow method which is among the many different heuristics for choosing a suitable $K$. The way it works is we run the algorithm using different values of $K$ and plot the heterogeneity. It operates in such a way that, for different values of $K$ the heterogeneity is plotted. In general, this measurement decreases when the value of $K$ increases since the size of the clusters decreases. The point where this measurement starts to flat out (elbow on the plot) corresponds to the optimal value of $K$ [25].

*2.5. Design of Layered Hidden Markov Model (LHMM) based Intrusion Detection System (IDS)*

This section discusses the novel layered HMM which is designed to detect multi-stage attacks. This layering technique can be further extended beyond the specific structure (Figure 3) which is simulated and discussed in this work.

2.5.1. Problem Statement Summary and Preliminaries

(1)    The problem space is decomposed into two separate layers of HMMs.
(2)    Each layer constitutes two levels: the observation data is used to train the HMMs and estimate the model's parameters at the first level of each layer and those parameters are used to find the most probable sequence of hidden states at the second level of the same layer.
(3)    The probable observable state sequences from each of the HMMs are used to construct the training data at Layer 2. It will be used for training the upper layer HMM which will be able to use the information from the lower layer HMMs to learn new patterns which are not possibly recognized by the lower layer HMMs.

2.5.2. Model Structure of HMM

An HMM is a double stochastic process. In other words, it represents two related stochastic process with the underlying stochastic process that is not necessarily observable but can be observed by another set of stochastic processes that produces the sequence of observations [26,27].

A typical notation for a discrete-observation HMM is

T = observation sequence length
$N$ = number of states in the model
$M$ = number of distinct observation symbols per state

$Q = \{q_1, q_2, \ldots, q_N\}$ = distinct "hidden" states of the Markov process
$V = \{v_1, v_2, \ldots, v_M\}$ = set of observation symbols per state
$S = \{s_1, s_2, \ldots, s_N\}$ = the individual states

It is specified by a set of parameters (A, B, Π) and each of the parameters is described below. At time $t$, $o_t$ and $q_t$ denote the observation and state symbols respectively.

1.    The prior (initial state) distribution $\Pi = \Pi_i$ where $\Pi_i = P(q_1 = s_i)$ are the probabilities of $s_i$ being the first state in a state sequence.
2.    The probability of state transition matrix $A = \{a_{ij}\}$ where $a_{ij} = P(q_{t+1} = s_j | q_t = s_i)$, is the probability of going from state $s_i$ to state $s_j$.
3.    The observation (emission) transition probability distribution $B = \{b_{ik}\}$ where $b_i(k) = P(o_t = v_j | q_t = s_i)$ is the probability of observing state $s_k$ given $q_t = s_i$.

Conventionally, the HMM model is represented by $\lambda = (A, B, \Pi)$. Given an HMM model, there are three problems to solve. One of the problems, also known as model training, is adjusting the model parameters to maximize the probability of the observation given a particular model and this is achieved using Baum-Welch algorithm which is a type of Expectation Maximization (EM) [28,29]. This procedure computes the maximum-likelihood estimates, local maxima, of the HMM model parameters (A, B, Π) using the forward and backward algorithms. In another word, for HMM models, $\lambda_1, \lambda_2, \ldots, \lambda_n$ and a given sequence of observations, $O = o_1, o_2, \ldots, o_t$, we choose $\lambda = (A, B, \Pi)$ such that $P(O|\lambda_i)$, $i = 1, 2, \ldots, n$ is locally maximized.

2.5.3. Model Structure of Two-Layered HMM (LHMM)

1.    At Layer 1, we have $HMM_1, HMM_2, \ldots, HMM_p$ with their corresponding number of hidden states $S_1, S_2, \ldots, S_p$.

2. Considering the same time granularity ($t = T$) of each of the HMMs,

- The observation sequence for each of the HMMs are given as:

$$O_1^T = \{O_1^1, O_1^2, \ldots, O_1^T\}, O_2^T = \{O_2^1, O_2^2, \ldots, O_2^T\}, \ldots, O_p^T = \{O_p^1, O_p^2, \ldots, O_p^T\}$$

- The probable sequence of states for each of the HMMs are given as:

$$Q_1^T = \{q_1^1, q_1^2, \ldots, q_1^T\}, \{q_2^1, q_2^2, \ldots, q_2^T\}, \ldots, \{q_p^1, q_p^2, \ldots, q_p^T\}$$

3. A new feature vector is constructed from the Layer 1 HMMs probable sequence of states. This statistical feature can be considered as a new data matrix where VQ can be applied and a new sequence of observations will be created from the Layer 2 HMM.

The feature vector is constructed as follows:

$$f_i = \begin{pmatrix} q_1^i \\ \vdots \\ q_T^i \end{pmatrix}, \ \forall i = 1, 2, \ldots, p \tag{5}$$

$$F = (f_1, f_2, \ldots f_j), \ \forall j = 1, 2, \ldots, p \tag{6}$$

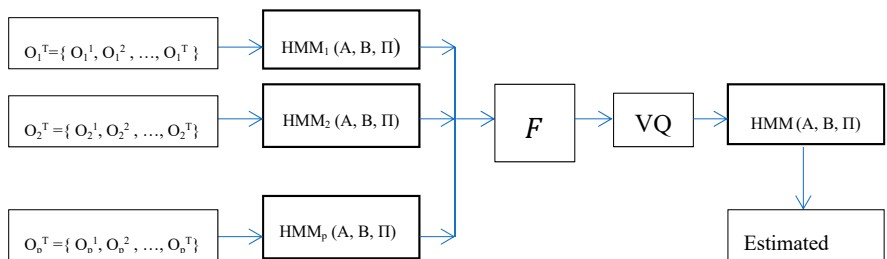

**Figure 3.** Multi-Layered HMM (LHMM) model structure.

### 2.5.4. Learning in LHMM

The models at Layer 1 and Layer 2 are trained independently to HMM model at Layer 2. Every HMM in each layer constitutes two levels where the first level determines model parameters and the second level finds the most probable sequence of states. At the first level of Layer 1, given the discrete time sequence of observations, the Baum-Welch algorithm is used for training the outputs of probable sequences. At the second level of Layer 1 and Layer 2, the Viterbi algorithm is used for finding the probable sequences based on the output of learned parameters from first level of same layer.

1. Learning at Layer 1

- Vector Quantization technique using K-Means Clustering applied on the training dataset.
- The Baum-Welch, an Expectation Maximization (EM), is used to compute the maximum log-likelihood estimates of the HMM model parameters (A, B, Π).

2. Learning at Layer 2

- Vector Quantization technique using K-Means Clustering, as defined in Section 2.4.2, is applied on the training dataset. Here the training dataset corresponds to matrix F in Equation (6).
- As we have a single HMM at Layer 2, the Baum-Welch method is used to compute the maximum log-likelihood estimates of the HMM model parameters (A, B, Π).

For the simulation of the LHMM, the two HMMs considered at the lower layer are HTTP and SSH traffic.

## 3. Results

This section discusses the results of the data processing for the modeled IDS: the dataset analysis using PCA for dimension reduction, K-Means clustering for vector quantization and finally the results of the LHMM.

### 3.1. Data Processing Results

1. **HTTP** traffic PCA Analysis

The results of HTTP traffic PCA analysis are demonstrated below. Figure 4 shows the scree plot of the dimensions with respect to percentage of explained variance and eigenvalues. The elbow method can be used to determine the number of dimensions to be retained. Equivalently, the cumulative percent of variance with respect to the number of dimensions, as shown in Table 1, can be used to determine the number of dimensions to be retained.

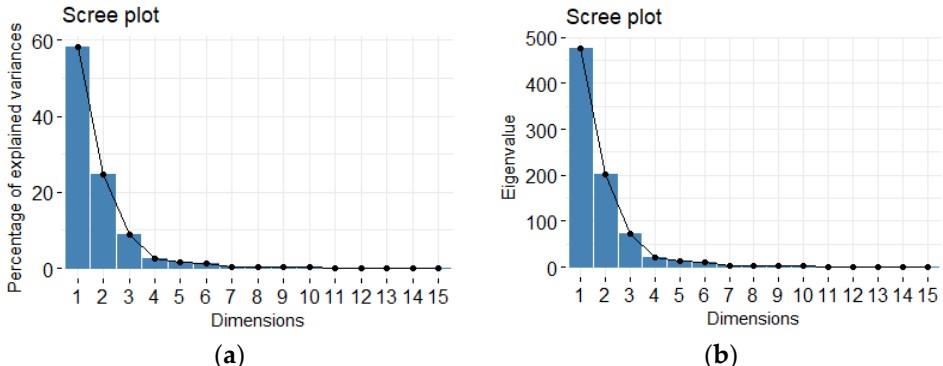

(**a**)　　　　　　　　　　　　　(**b**)

**Figure 4.** (**a**) Scree Plot of the percentage of explained variance; (**b**) Scree plot of eigenvalues.

**Table 1.** Principal components with their variance contribution.

|  | Eigenvalue | Percent of Variance | Cumulative Percent of Variance |
|---|---|---|---|
| Dim.1 | $4.762758 \times 10^2$ | $5.826067 \times 10^1$ | 58.26067 |
| Dim.2 | $2.011200 \times 10^2$ | $2.460210 \times 10^1$ | 82.86276 |
| Dim.3 | $7.160326 \times 10^1$ | 8.758904 | 91.62167 |
| Dim.4 | $2.027640 \times 10^1$ | 2.480320 | 94.10199 |
| Dim.5 | $1.437619 \times 10^1$ | 1.758575 | 95.86056 |
| Dim.6 | $1.010701 \times 10^1$ | 1.236345 | 97.09691 |
| Dim.7 | 4.111222 | $5.029072 \times 10^{-1}$ | 97.59981 |
| Dim.8 | 4.057777 | $4.963695 \times 10^{-1}$ | 98.09618 |
| Dim.9 | 2.812110 | $3.439927 \times 10^{-1}$ | 98.44018 |
| Dim.10 | 2.672606 | $3.269278 \times 10^{-1}$ | 98.7671 |
| Dim.11 | 1.433143 | $1.753099 \times 10^{-1}$ | 98.94241 |
| Dim.12 | 1.283170 | $1.569643 \times 10^{-1}$ | 99.09938 |
| Dim.13 | 1.226318 | $1.500099 \times 10^{-1}$ | 99.24939 |
| Dim.14 | $9.864789 \times 10^{-1}$ | $1.206715 \times 10^{-1}$ | 99.37006 |
| Dim.15 | $8.110305 \times 10^{-1}$ | $9.920970 \times 10^{-2}$ | 99.46927 |
| Dim.16 | $7.242188 \times 10^{-1}$ | $8.859041 \times 10^{-2}$ | 99.55786 |
| Dim.17 | $6.931104 \times 10^{-1}$ | $8.478506 \times 10^{-2}$ | 99.64264 |
| Dim.18 | $6.044243 \times 10^{-1}$ | $7.393649 \times 10^{-2}$ | 99.71658 |
| Dim.19 | $4.243713 \times 10^{-1}$ | $5.191142 \times 10^{-2}$ | 99.76849 |

For the HTTP traffic 8 principal components which correspond to 98.09618% of the explained variance are selected from Table 1. Those selected 8 PCs and the first 6 (head) out of the total number of features are shown in Table 2. Each PC is constructed from a linear combination of the total features and their multiplying coefficients specified the table.

**Table 2.** The selected 8 Principal components and head (6 of the original features only displayed here).

| | PC1 | PC2 | PC3 | PC4 | PC5 | PC6 | PC7 | PC8 |
|---|---|---|---|---|---|---|---|---|
| Flow_ID | $7.955283 \times 10^{-5}$ | $1.457065 \times 10^{-2}$ | $1.644281 \times 10^{-3}$ | $3.287887 \times 10^{-1}$ | $5.998037 \times 10^{-2}$ | $9.81543 \times 10^{-2}$ | $2.131072 \times 10^{-1}$ | $3.108175 \times 10^{-1}$ |
| Source_IP | $7.387470 \times 10^{-3}$ | $4.617471 \times 10^{-2}$ | $4.837233 \times 10^{-3}$ | $2.641242 \times 10^{-1}$ | $2.070467 \times 10^{-2}$ | $2.057702 \times 10^{-2}$ | $2.77837 \times 10^{-30}$ | $6.927052 \times 10^{-1}$ |
| Destination_IP | $2.773583 \times 10^{-7}$ | $1.824271 \times 10^{-2}$ | $7.112974 \times 10^{-3}$ | $4.124236 \times 10^{-1}$ | $2.091758 \times 10^{-4}$ | $5.842059 \times 10^{-2}$ | $1.562471 \times 10^{-1}$ | $2.848223 \times 10^{-1}$ |
| Destination_Port | $1.232595 \times 10^{-30}$ | $1.232595 \times 10^{-3}$ | $5.772448 \times 10^{-3}$ | $1.774937 \times 10^{-28}$ | $8.493351 \times 10^{-30}$ | $2.097338 \times 10^{-29}$ | $1.203706 \times 10^{-29}$ | $4.930381 \times 10^{-30}$ |
| Flow.Duration | $1.309048$ | $4.731603$ | $2.933845 \times 10^{-4}$ | $3.366436 \times 10^{-1}$ | $4.514761$ | $5.269066$ | $9.913239$ | $6.871774 \times 10^{-1}$ |
| Total.Fwd.Packets | $8.387569 \times 10^{-2}$ | $2.23493 \times 10^{-3}$ | $2.357599 \times 10^{-2}$ | $3.299392 \times 10^{-3}$ | $3.453238 \times 10^{-2}$ | $1.038723 \times 10^{-1}$ | $6.913426 \times 10^{-1}$ | $4.477584$ |

2. **SSH** Traffic PCA Analysis

The results of SSH traffic PCA analysis are demonstrated below. Figure 5 shows the scree plot of the dimensions with respect to percentage of explained variance and eigenvalues. The elbow method can be used to determine the number of dimensions to retain. Equivalently, the cumulative percent of variance with respect to the number of dimensions, as shown in Table 3, can be used to determine the number of dimensions to be retained.

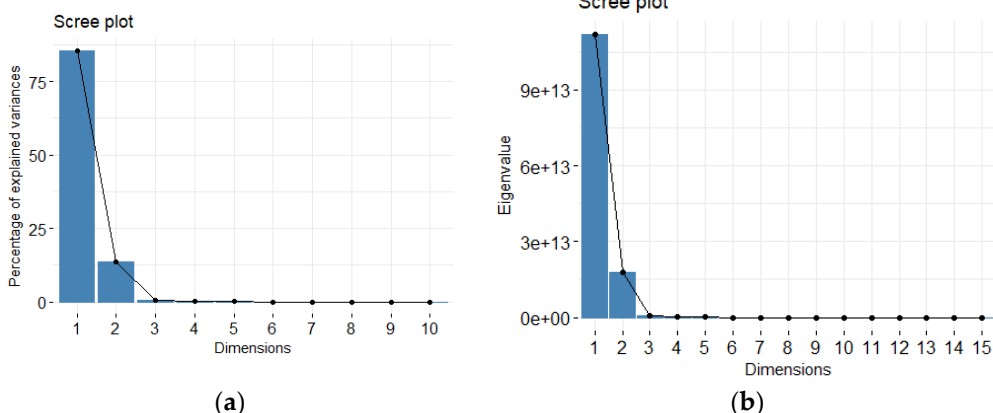

**Figure 5.** (**a**) Scree Plot of the percentage of explained variance; (**b**) Scree plot of eigenvalues.

**Table 3.** Principal components with their variance contribution.

|  | Eigenvalue | Percent of Variance | Cumulative Percent of Variance |
|---|---|---|---|
| Dim.1 | $1.11834 \times 10^{14}$ | $8.53962 \times 10^{1}$ | 85.39622 |
| Dim.2 | $1.77023 \times 10^{13}$ | $1.35175 \times 10^{1}$ | 98.91367 |
| Dim.3 | $6.84686 \times 10^{11}$ | $5.22826 \times 10^{-1}$ | 99.4365 |
| Dim.4 | $4.27484 \times 10^{11}$ | $3.26427 \times 10^{-1}$ | 99.76292 |
| Dim.5 | $2.53906 \times 10^{11}$ | $1.93883 \times 10^{-1}$ | 99.95681 |
| Dim.6 | $2.83901 \times 10^{10}$ | $2.16787 \times 10^{-2}$ | 99.97848 |
| Dim.7 | $1.66896 \times 10^{10}$ | $1.27442 \times 10^{-2}$ | 99.99123 |
| Dim.8 | $6.49554 \times 10^{9}$ | $4.95999 \times 10^{-3}$ | 99.99619 |
| Dim.9 | $3.08202 \times 10^{9}$ | $2.35343 \times 10^{-3}$ | 99.99854 |
| Dim.10 | $1.31229 \times 10^{9}$ | $1.00207 \times 10^{-3}$ | 99.99954 |
| Dim.11 | $3.37909 \times 10^{8}$ | $2.58027 \times 10^{-4}$ | 99.9998 |
| Dim.12 | $1.19652 \times 10^{8}$ | $9.13658 \times 10^{-5}$ | 99.99989 |
| Dim.13 | $5.03480 \times 10^{7}$ | $3.84457 \times 10^{-5}$ | 99.99993 |
| Dim.14 | $3.36857 \times 10^{7}$ | $2.57224 \times 10^{-5}$ | 99.99996 |
| Dim.15 | $2.31915 \times 10^{7}$ | $1.77090 \times 10^{-5}$ | 99.99998 |
| Dim.16 | $1.72495 \times 10^{7}$ | $1.31717 \times 10^{-5}$ | 99.99999 |
| Dim.17 | $1.00092 \times 10^{7}$ | $7.64299 \times 10^{-6}$ | 100 |

For the SSH traffic, 4 principal components which correspond to 99.76292% of the explained variance are selected from Table 3. The first three dimensions of the PCA retains slightly over 99% of the total variance (i.e., information) contained in the data. Those selected 3 PCs and the first 6 (head) out of the total number of features are shown in Table 4.

**Table 4.** The selected 4 Principal components and head (6 of the original features only displayed).

| | PC1 | PC2 | PC3 | PC4 |
|---|---|---|---|---|
| **Flow_ID** | $5.20233 \times 10^{-10}$ | $9.18848 \times 10^{-9}$ | $4.46286 \times 10^{-8}$ | $9.02167 \times 10^{-8}$ |
| **Source_IP** | $6.00175 \times 10^{-13}$ | $1.93802 \times 10^{-12}$ | $1.48850 \times 10^{-11}$ | $1.77674 \times 10^{-11}$ |
| **Destination_IP** | $1.23260 \times 10^{-30}$ | $0.00000$ | $1.23260 \times 10^{-30}$ | $1.01882 \times 10^{-29}$ |
| **Destination_Port** | $0.00000$ | $0.00000$ | $0.00000$ | $1.23260 \times 10^{-30}$ |
| **Flow.Duration** | $3.52624 \times 10^{1}$ | $7.10929 \times 10^{-1}$ | $8.77584 \times 10^{-1}$ | $7.02152 \times 10^{-2}$ |
| **Total.Fwd.Packets** | $1.22041 \times 10^{-11}$ | $5.02931 \times 10^{-11}$ | $6.34210 \times 10^{-9}$ | $4.41869 \times 10^{-10}$ |

*Vector Quantization*

The goal of the vector quantization is to simplify the dataset from a complex higher dimensional space into a lower dimensional space so that it can be easier for visualization and find patterns. In this work, it is achieved by using a very common unsupervised machine learning algorithm such as K-Means clustering.

To determine the number of clusters (K) in K-Means, the simplest method involves plotting the number of clusters against the within groups sum of squares and find pick the 'elbow' point in this plot. This is similar in concept with the scree plot for the PCA discussed in Section 2.3.

1.  **HTTP** Clustering Analysis

K-Means clustering is applied on the HTTP traffic after PCA and the number of clusters is determined where the elbow occurs in Figure 6 which is K = 4. The plot shows the within cluster sum of squares (wcss) as the number of clusters (K) varies.

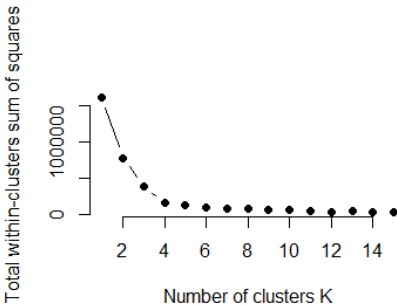

**Figure 6.** The wcss vs. number of clusters for HTTP traffic.

2.  **SSH** Clustering Analysis

On the SSH traffic, K-Means is applied after PCA analysis similar to the HTTP traffic above and the number of clusters where the elbow occurs, K = 3, is selected from Figure 7.

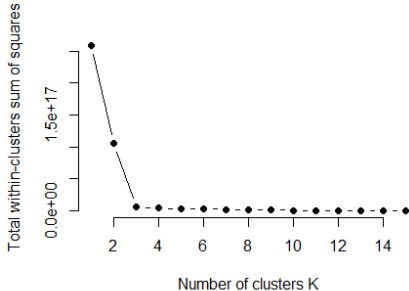

**Figure 7.** The wcss vs. number of clusters for SSH traffic.

### 3.2. Layered HMM Experimental Results

For the simulation of the LHMM, the two HMMs considered the lower layer are HTTP and SSH traffic.

### 3.2.1. Training the LHMM

The lower layer HMMs are trained using their corresponding training data and the optimized model parameters are determined using the Baum-Welch algorithm.

1.  **HTTP HMM training**

    **HMM model parameters** (A, B, Π) after training are as shown below.

$$A = \begin{bmatrix} 0.9827 & 0.0173 \\ 0.0140 & 0.9860 \end{bmatrix}$$

$$B = \begin{bmatrix} 0.3088 & 0.0973 & 0.2007 & 0.3932 \\ 0.0000 & 0.8129 & 0.0952 & 0.0919 \end{bmatrix}$$

$$\pi = \begin{bmatrix} 1 \\ 0 \end{bmatrix}$$

The number of hidden states in the HTTP training traffic is shown in Table 5. The corresponding state symbols sequence is plotted against the HTTP training data in Figure 8.

**Table 5.** Hidden State Symbols of the HTTP traffic.

| State Symbols | HTTP |
| :---: | :---: |
| 1 | HTTP-BENIGN |
| 2 | HTTP-Web-attack-bruteforce |

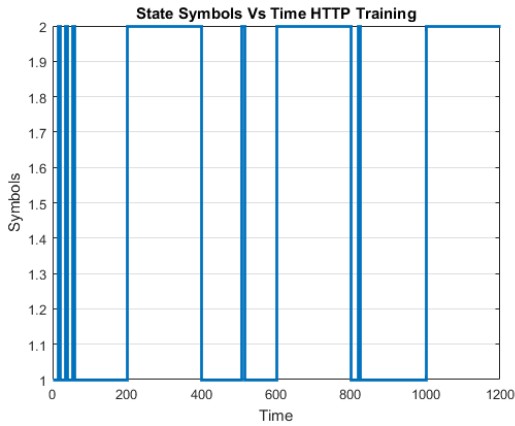

**Figure 8.** State symbols against the time series HTTP training data.

2.  **Training the SSH HMM**

    **HMM model parameters** (A, B, Π) after training were found to be

$$A = \begin{bmatrix} 0.9772 & 0.0228 \\ 0.0308 & 0.9692 \end{bmatrix}$$

$$B = \begin{bmatrix} 0.6135 & 0.1518 & 0.2348 \\ 0.5092 & 0.2482 & 0.2427 \end{bmatrix}$$

$$\pi = \begin{bmatrix} 1 \\ 0 \end{bmatrix}$$

The number of hidden states in the SSH training traffic is shown in Table 6. The corresponding state symbols sequence is plotted against the SSH training data in Figure 9.

**Table 6.** Hidden State Symbols of the SSH traffic.

| State Symbols | SSH |
|:---:|:---:|
| 1 | SSH-BENIGN |
| 2 | SSH-Patator |

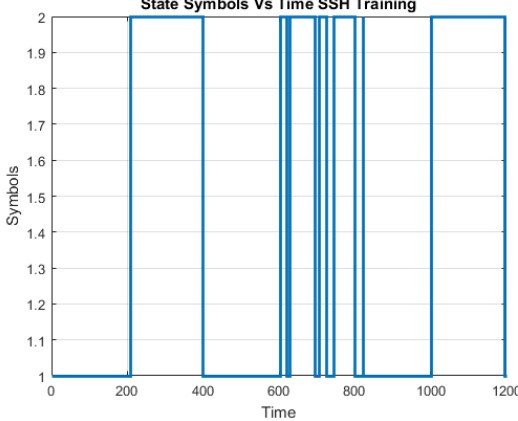

**Figure 9.** State symbols against the time series SSH training data.

## 3. Training Upper Layer HMM

Following the training the lower layer HMMs, the model parameters (A, B, Π) after training of the upper layer HMM were found to be

$$A = \begin{bmatrix} 0.9860 & 0.0000 & 0.0000 & 0.0000 & 0.0140 \\ 0.0011 & 0.8962 & 0.0026 & 0.0988 & 0.0014 \\ 0.0000 & 0.2557 & 0.2870 & 0.4240 & 0.0334 \\ 0.0255 & 0.7642 & 0.0024 & 0.1401 & 0.0678 \\ 0.0568 & 0.0105 & 0.0684 & 0.0007 & 0.8636 \end{bmatrix}$$

$$B = \begin{bmatrix} 0.0000 & 1.0000 & 0.0000 & 0.0000 \\ 1.0000 & 0.0000 & 0.0000 & 0.0000 \\ 0.9996 & 0.0000 & 0.0004 & 0.0000 \\ 1.0000 & 0.0000 & 0.0000 & 0.0000 \\ 0.0000 & 0.0000 & 1.0000 & 0.0000 \end{bmatrix}$$

$$\pi = \begin{bmatrix} 1 \\ 0 \\ 0 \\ 0 \\ 0 \end{bmatrix}$$

The hidden states in the upper layer HMM training traffic is shown in Table 7. The corresponding state symbols sequence is plotted against the upper layer HMM training data in Figure 10.

**Table 7.** Hidden State Symbols of the Upper layer in the training data.

| State Symbols | HTTP | SSH |
|---|---|---|
| 1 | HTTP-BENIGN | SSH-BENIGN |
| 2 | HTTP-Web-attack-bruteforce | SSH-Patator |
| 3 | HTTP-Web-attack-bruteforce | SSH-BENIGN |
| 4 | HTTP-Web-attack-bruteforce | SSH-Patator |
| 5 | HTTP-BENIGN | SSH-Patator |

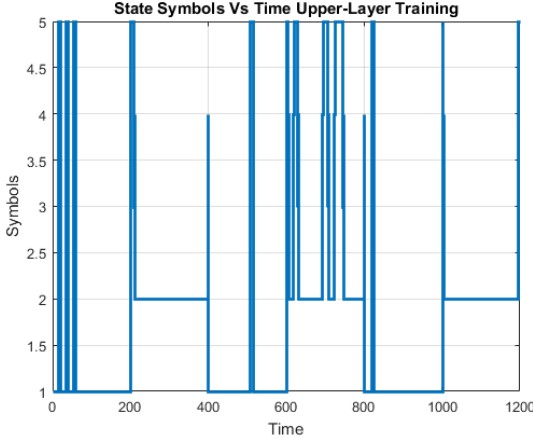

**Figure 10.** State symbols against the time series Upper training data.

### 3.2.2. Testing the LHMM

The sequence of network states during testing are determined using the Viterbi algorithm which uses as an input the model parameters that are determined during training phase.

### 1. Testing HTTP HMM

During testing, the hidden states of the HTTP HMM shown in Table 8 are similar to the training phase hidden states. The corresponding sate symbol sequences are plotted with the HTTP testing data in Figure 11.

**Table 8.** Hidden State Symbols of the HTTP traffic during testing.

| State Symbols | HTTP |
|---|---|
| 1 | HTTP-BENIGN |
| 2 | HTTP-Web-attack-bruteforce |

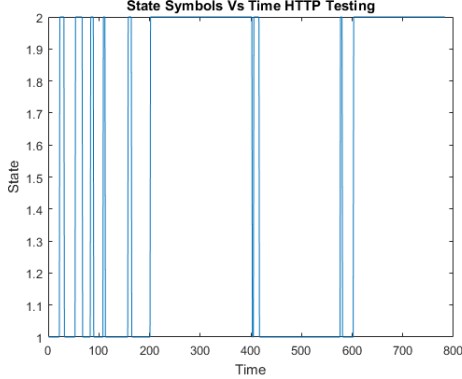

**Figure 11.** State symbols against the time series HTTP test data.

## 2. Testing SSH HMM

Similarly, the SSH HMM hidden state symbols are the same as the SSH training data hidden states as shown in Table 9. These state symbols are plotted against a time series data of the testing data as shown in Figure 12.

**Table 9.** Hidden State Symbols of the SSH traffic in testing data.

| State Symbols | SSH |
|:---:|:---:|
| 1 | SSH-BENIGN |
| 2 | SSH-Patator |

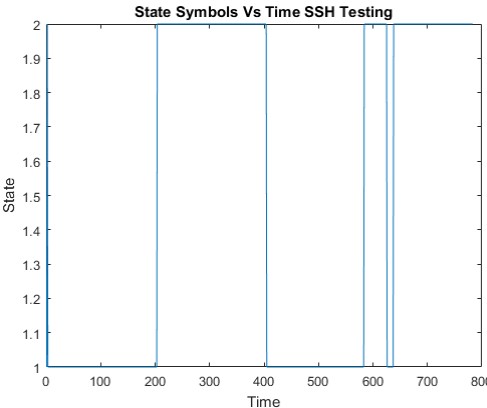

**Figure 12.** State symbols against the time series SSH test data.

## 3. Testing Upper Layer HMM

The Upper layer HMM testing hidden states are shown in Table 10 and constitute the hidden states of the lower layer HMMs. And final results shown in Figure 13 prove the validity of the multi-layer HMM in determining the hidden states within the IDS detection engine.

**Table 10.** Hidden State Symbols of the Upper layer testing data.

| State Symbols | HTTP | SSH |
|:---:|:---:|:---:|
| 1 | HTTP-BENIGN | SSH-BENIGN |
| 2 | HTTP-Web-attack-bruteforce | SSH-Patator |
| 3 | HTTP-Web-attack-bruteforce | SSH-BENIGN |
| 4 | HTTP-Web-attack-bruteforce | SSH-Patator |
| 5 | HTTP-BENIGN | SSH-Patator |

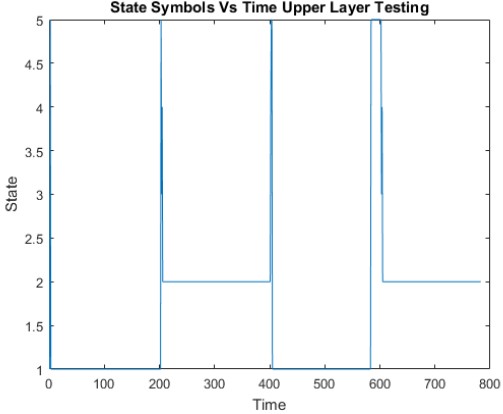

**Figure 13.** State symbols against the time series Upper test data.

## 4. Discussion

A high detection rate is essential in a machine learning based IDS alongside the evaluation metrics aforementioned. The main aspects to consider when measuring the accuracy are

- True Positive (TP): Number of intrusions correctly detected.
- True Negative (TN): Number of non-intrusions correctly detected.
- False Positive (FP): Number of non-intrusions incorrectly detected.
- False Negative (FN): Number of intrusions incorrectly detected.

The performance of the LHMM by can be evaluated by calculating the common performance measures: Accuracy, Sensitivity, Specificity, Precision, Recall and F-Measure. Those metrics are among the few considered which are commonly used for evaluating IDS performance [30–32].

a)  Accuracy: the proportion of true results (both true negatives and true positives) with respect to the total number.

$$Accuracy = \frac{t_p + t_n}{t_p + t_n + f_p + f_n}$$

b)  Precision: the fraction of the states which were classified as the interesting state (loaded in this case) that are really that state.

$$Precision = \frac{t_p}{t_p + f_p}$$

c)  Recall: the fraction of the interesting states that were correctly predicted as such. It is also referred as Sensitivity.

$$Recall = \frac{t_p}{t_p + f_n}$$

d)  F-measure: a combination of precision and recall and provides the percentage of positively classified incidents that are truly positive.

$$F1 = \frac{2x Precision \times Recall}{Precision + Recall}$$

The performance of the LHMM for the above test is as follows:

[Accuracy Precision Recall f_measure] = [0.9898 0.9793 1.0000 0.9895]

Additionally, the CPU consumption, the throughput and the power consumption are important metrics for the evaluation of intrusion detection systems running on different hardware on specific settings such as high-speed networks, or on hardware with limited resources. But in this work the performance of the developed IDS is measured solely on the above metrics.

As compared to a single-layer HMM, the layered approach has several advantages: (1) A single-layer HMM possibly have to be trained on a large number of observations space. In this case, the model can be over-fitted when enough training data is not used. As the observation space increases, the amount of data needed to train the model well also increases. As a result it incurs what is commonly referred as the curse of dimensionality. On the contrary, the layers in this approach are trained over small-dimensional observation spaces which results in more stable models and do not require large amount of training data. (2) The lower layer HMMs are defined and trained with their corresponding data as needed. (3) The second layer-HMM is less sensitive to variations in the lower layer features as the observations are the outputs from each lower layer HMMs, which are expected to be well trained. (4) The two layers (lower and upper) are expected to be well trained independently. Thus, we can explore different HMM combination systems. In particular, we can replace the first layers

HMMs with models that are more suitable for network traffic data sequences, with the goal of gaining understanding of the nature of the data being used. The framework is thus easier to improve and interpret individually at each level. (5) The layered framework in general can be expanded to learn new network traffics that can be defined in the future by adding additional HMMs in the lower layers.

The results demonstrate how a Markov Model can capture the statistical behavior of a network and determine the presence of attacks and anomalies based on known normal network behaviors gathered from training data. Using the vector quantization method, we are able to include multiple dimensions of information into the model and this will be helpful in reducing false positives and determining more attack states in the future. The model can be re-trained to identify new network states based on the robustness of the training data. This is a promising approach because it is extensible.

## 5. Conclusions

This work highlights the potential for use of multi-layer HMM-based IDS. PCA using SVD is used for dimension reduction of CICIDS2017 data after feature selection and feature creation. By applying K-Means clustering as a vector quantization technique on the reduced data, the cluster labels are used as a sequence of observation to an HMM model. The IDS is based on multi-layer HMM.

The proposed approach has been developed and verified to resolve the common flaws in the application of HMM to IDS commonly referred as the curse of dimensionality. It factors a huge problem of immense dimensionality to a discrete set of manageable and reliable elements. The multi-layer approach can be expanded beyond 2 layers to capture multi-phase attacks over longer spans of time. A pyramid of HMMs can resolve disparate digital events and signatures across protocols and platforms to actionable information where lower layers identify discrete events (such as network scan) and higher layers new states which are the result of multi-phase events of the lower layers. The concepts of this approach have been developed but the full potential has not been demonstrated.

**Author Contributions:** The authors contributed equally to this work.

**Funding:** This research received no external funding.

**Conflicts of Interest:** The authors declare no conflict of interest.

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
