# Peer review of "Multi-Layer Hidden Markov Model Based Intrusion Detection System"

_make, doi:10.3390/make1010017_

Reviewer 1 Report

The paper is well written and present a Multi-Layer Hidden Markov Model for intrusion detection systems.

Within Section 2.1 CICIDS2017 the manuscript highlights the dataset being used, I am however surprised not to see a paragraph on the choice of the dataset and the strength and weaknesses of other datasets. Numerous authors such as  

Al Tobi and Duncan KDD 1999 generation faults: a review and analysis

and

Hindi et al. A Taxonomy and Survey of Intrusion Detection System Design Techniques, Network Threats and Datasets

Have highlighted the problems of other datasets and their limitation. I believe such a discussion would strengthen the CICDS2017 choice. 

Figure 2 should be better explained. 

2.4.1 problem statem seem to arrive late in the text, this should be stated upfront, to ease the readability of the manuscript. 

Could Figure 5 & 6 not be replaced by tables, rather than print screens ? 

The accuracy is a measure that is often not reflecting the true value of a model I.e. 90% valid traffic with 10% malicious traffic can still provide 90% accuracy.  Could you elaborate on the choices made, or provide another measure / justification.

Author Response

Please find attached the minor revisions done on the updated manuscript

Reviewer 2 Report

The manuscript describes an application of multi-layer hidden Markov model for intrusion detection in the network traffic. In fact, the proposal is not novel (multiple articles presents exactly this method), but the paper it is well described, the method was verified over quite new dataset (CICIDS2017 dataset) and the provided results emphasize the utilization of this method. Thus, I propose this paper for publication.

Author Response

The following minor revisions are done on the updated manuscript

·         Section 2.1 CICIDS2017 Data Processing updated (please have a look at of the updated manuscript)

·         Figure 2 is further elaborated ( please have a look at of the updated manuscript) how the data matrix is decomposed using the value form equation of SVD 

·         Figure 5 & 6 and the other print screens are also replaced by tables.  Table numbers and figure numbers are updated accordingly.